# A room temperature continuous-wave nanolaser using colloidal quantum wells

Zhili Yang[1], Matthew Pelton[2], Igor Fedin[3], Dmitri V. Talapin [3] & Edo Waks[1,4]

Colloidal semiconductor nanocrystals have emerged as promising active materials for solution-processable optoelectronic and light-emitting devices. In particular, the development of nanocrystal lasers is currently experiencing rapid progress. However, these lasers require large pump powers, and realizing an efficient low-power nanocrystal laser has remained a difficult challenge. Here, we demonstrate a nanolaser using colloidal nanocrystals that exhibits a threshold input power of less than 1 μW, a very low threshold for any laser using colloidal emitters. We use CdSe/CdS core-shell nanoplatelets, which are efficient nanocrystal emitters with the electronic structure of quantum wells, coupled to a photonic-crystal nanobeam cavity that attains high coupling efficiencies. The device achieves stable continuous-wave lasing at room temperature, which is essential for many photonic and optoelectronic applications. Our results show that colloidal nanocrystals are suitable for compact and efficient optoelectronic devices based on versatile and inexpensive solution-processable materials.

[1] Department of Electrical Engineering and Institute for Research in Electronics and Applied Physics, University of Maryland, College Park, Maryland 20742, USA. [2] Department of Physics, University of Maryland, Baltimore County, Baltimore, Maryland 21250, USA. [3] Department of Chemistry and James Franck Institute, University of Chicago, Chicago, Illinois 60637, USA. [4] Joint Quantum Institute, University of Maryland and National Institute of Standards and Technology, College Park, Maryland 20742, USA. Correspondence and requests for materials should be addressed to M.P. (email: mpelton@umbc.edu) or to E.W. (email: edowaks@umd.edu)

Solution-processed emitters offer a compelling gain material for laser applications. These materials can be synthesized using inexpensive colloidal chemistry techniques, alleviating the need for complex epitaxial deposition methods. They can also be placed on a broad range of substrates using convenient solution-deposition techniques. For example, fluorescent molecules embedded in polymer can act as a gain material for lasers[1]. However, these emitters typically photo-bleach on short timescales of seconds to minutes[2], and also suffer from rapid gain quenching due to intersystem crossing into dark triplet states, making continuous-wave operation challenging[1]. Colloidally synthesized quantum wires have also shown lasing with the possibility of electrical injection[3]. However, the optical mode in nanowire lasers extends for several microns along the wire axis, resulting in high threshold pump powers[4].

Semiconductor nanocrystals synthesized by colloidal chemistry have emerged as another promising solution-processable gain medium[5]. They exhibit significantly better photostability than fluorescent molecules[6] and do not suffer from gain quenching due to intersystem crossing. Perhaps the most well-studied nanocrystal emitters are quantum dots, which have historically suffered from rapid Auger recombination that made lasing difficult to achieve[7]. However, recent advances in quantum-dot

synthesis have largely mitigated this problem and enabled amplified spontaneous emission and lasing in large-mode-volume cavities such as Fabry–Pérot, distributed Bragg reflector, and microsphere cavity resonators over the entire visible spectral range[8–13] and extending out to infrared frequencies[14]. In addition to quantum dots, new colloidal materials such as nanorods[15, 16], nanoplatelets[17–20] and perovskite nanocrystals[21, 22] have emerged as good gain materials for room temperature amplified spontaneous emission and lasing. However, all of these past works required large pump powers to achieve lasing threshold due to their large cavity mode volume and high loss.

One way to significantly reduce these large threshold powers is to employ small mode-volume cavities that reduce the size of the active material and enhance the spontaneous emission coupling efficiency[23]. Nanolasers based on various materials have been previously demonstrated using both dielectric cavities[24] and metallic nanostructures[25]. However, the development of nano-lasers using solution-processable semiconductor nanocrystals has proven difficult. A number of works incorporated colloidal quantum dots into nanocavities[26–28], but were unable to reach lasing due to rapid Auger recombination, which resulted in gain quenching. The poor uniformity of quantum dot films deposited on nanophotonic structures also posed a major

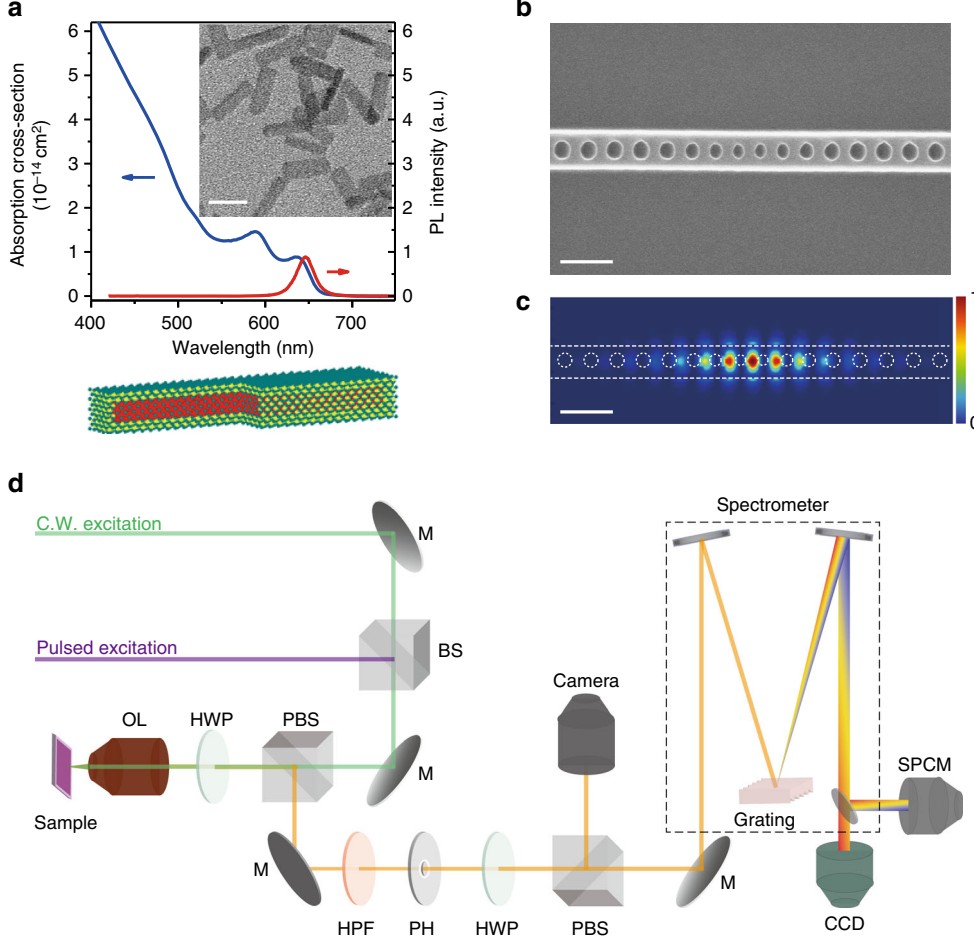

**Fig. 1** Design of the nanoplatelet nanolaser. **a** Absorption cross-section (*blue*) and photoluminescence (*red*) spectra of CdSe core / CdS shell nanoplatelets in solution. The inset shows a transmission-electron-microscope image of nanoplatelets, and the schematic below is an illustration of the nanoplatelet structure. Scale bar is 20 nm. **b** Scanning electron microscope image of a Silicon nitride photonic-crystal nanobeam cavity before deposition of the nanoplatelets. **c** Calculated electric-field intensity profile ($|E|^2$) of the fundamental mode in the cavity. Scale bars in both **b** and **c** are 500 nm. **d** Experimental setup. M: mirror, BS: beam spliter, PBS: polarizing beam spliter, HWP: half-wave plate, OL: objective lens, HPF: high-pass filter (cut-off at 600 nm), PH: pin hole as spatial fiter, SPCM: single photon counting module

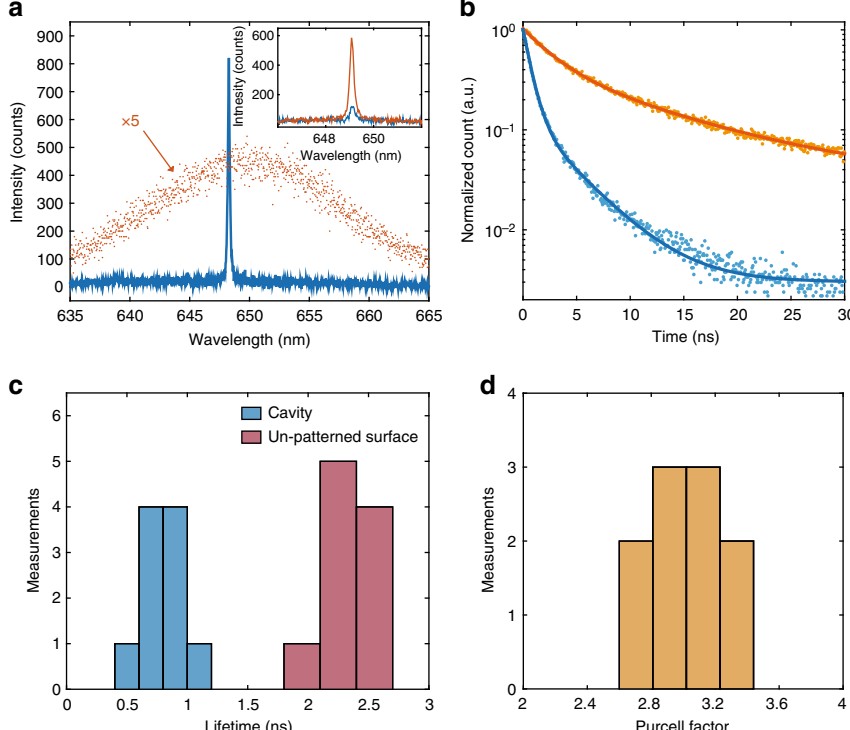

**Fig. 2** Emission modification and lifetime measurements. **a** Emission spectra from nanoplatelets on the cavity (*blue*) and in an un-patterned region (*orange*, multiplied by a factor of 5 for clarity), both measured under a CW excitation power of 10 µW. The inset shows emission spectra taken at polarization directions parallel (*blue*) and orthogonal (*orange*) to the nanobeam. **b** Normalized time-resolved photoluminescence from nanoplatelets on the cavity (*blue*) and in an un-patterned region (*orange*). Dots indicate the measured data, and the lines are fits to bi-exponential decay model with both fast and slow decay times of $\tau_1$ and $\tau_2$ as fit parameters. **c** Histogram of measured lifetimes for both the un-patterned regions (*purple*) and the cavities (*blue*). **d** Histogram of Purcell factors determined from the lifetime measurements in **c**

obstacle, and these films degraded rapidly under optical excitation[29]. Achieving a low-threshold nanolaser using such materials has therefore remained an outstanding challenge.

In this work, we report an experimental realization of a nanolaser using colloidally synthesized semiconductor nanocrystals. We use colloidal nanoplatelets, a new class of material, coupled to silicon-nitride nanobeam cavities to achieve continuous-wave room-temperature lasing. Colloidal nanoplatelets exhibit high gain and photostability, and also deposit as thin uniform films on top of the nanocavity, enabling us to overcome many of the challenges faced by previous efforts using quantum dots. We demonstrate lasing with a threshold input power of 0.97 µW, an extremely low threshold for any laser using colloidal emitters. We also estimate an absorbed power of 210 nW, which is equivalent to the lowest reported value for a room-temperature laser[30]. Our results are an important step towards efficient on-chip light emitters and photonic integrated devices based on colloidally synthesized solution-processable materials.

## Results

**Nanolaser components**. The gain material for our nanolaser is a thin film of colloidally synthesized CdS/CdSe/CdS nanoplatelets. These semiconductor nanoscale heterostructures confine carriers quantum-mechanically in one dimension, and are thus the colloidal analogue of epitaxially grown quantum wells[31]. Colloidal nanoplatelets are promising materials for laser applications because they exhibit lower Auger recombination rates compared to those in conventional colloidal quantum dots at equivalent exciton densities[32], and the core/shell structure significantly reduces emission intermittency (blinking) at a single nanoparticle level[33]. Thus, they can provide higher gain at equivalent

excitation powers. Figure 1a shows the absorption and emission spectrum of the nanoplatelets used in our experiment, along with a schematic illustration and a transmission electron microscope image. We provide a detailed description of the synthesis and characterization of the nanoplatelets in the Methods section.

In order to create a nanolaser, we couple the nanoplatelets to a silicon nitride photonic crystal nanobeam cavity[34]. Silicon nitride is an ideal substrate because it is highly transparent at the nanoplatelet emission wavelength and has a high index of refraction. It has the additional benefit that it can be fabricated with the same tools used for silicon electronic devices[35], opening opportunities for complementary metal-oxide-semiconductor (CMOS) process integration. Figures 1b, c show a scanning electron microscope image of the cavity structure along with the calculated mode profile obtained using finite-difference time-domain simulation. We provide full details of the device design and fabrication in the Methods section. Figure 1d illustrates the experimental setup to characterize the fabricated devices, and the Methods section describes the measurement procedure.

**Photoluminescence measurements**. Figure 2a shows emission spectra obtained by exciting the nanoplatelets at the cavity region, as well as a reference spectrum from a region on the un-patterned sample surface (multiplied by a factor of 5 for better visualization). In both cases, we use an excitation power of 10 µW. The spectrum of the un-patterned region exhibits a full-width half-maximum linewidth of 21 nm, which agrees with previously reported values for ensembles of nanoplatelets[31]. In contrast, the cavity emission shows a sharp resonance at the cavity centre frequency that is more than 8 times brighter than the peak emission from the un-patterned surface. We attribute this

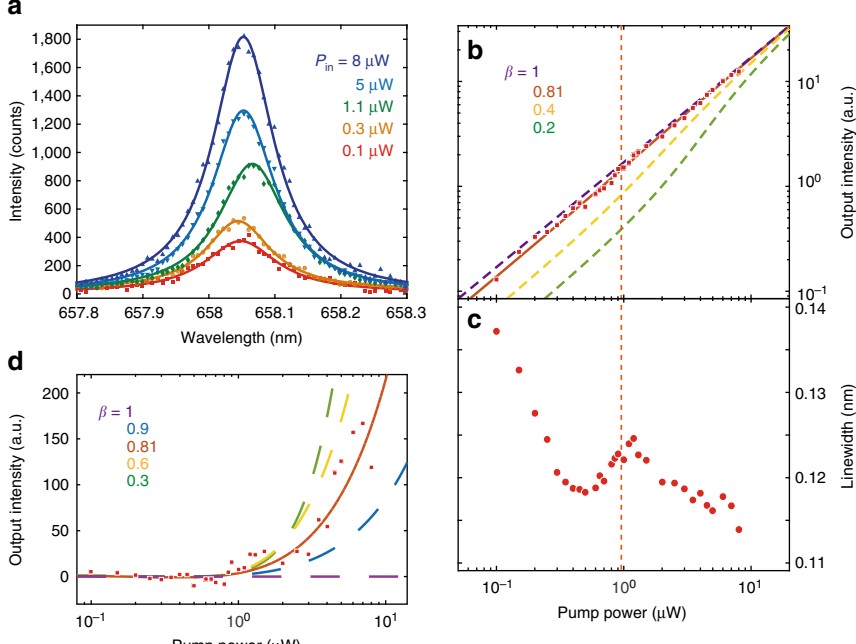

**Fig. 3** Observation and characterization of nanoplatelet lasing. **a** Emission spectra from nanoplatelets coupled to the cavity under different pump powers. Symbols indicate the measured data, and the lines are Lorentzian fits. Keys are pump powers for corresponding spectra in the same colours. **b** Output intensity, determined from the Lorentzian fits, as a function of the total pump power. *Red squares* are measured data, and the solid curve is a fit to the laser rate equation, corresponding to a spontaneous emission coupling efficiency $\beta = 0.81$. *Dashed curves* are calculation results for other values of $\beta$. **c** Cavity linewidth as a function of pump power. The *vertical dashed line* (*orange*) in both (**b**) and (**c**) indicates the lasing threshold obtained from the fit. **d** Output intensity after subtracting a linear fit to the data points below threshold. The solid curve is the fit to the laser rate equation, and *dashed curves* are calculated for other values of $\beta$

increase in brightness to both redirection of the nanoplatelet emission by the cavity as well as the Purcell effect (spontaneous emission enhancement into the cavity mode). We observe brighter intensity at the cavity resonance despite the fact that we excite many more nanoplatelets when looking at the bare surface as compared to the cavity, because the surface area of the nanocavity is much smaller than the surface area excited by the laser on the un-patterned surface. We calculate the surface area of the cavity, defined as the mode volume divided by the membrane thickness, to be $0.086\,\mu m^2$. In contrast, the illuminated surface area on the un-patterned surface is $0.45\,\mu m^2$ (see Methods).

The striking modification of the nanoplatelet emission spectrum is an indication of enhanced spontaneous emission from the nanoplatelets into the cavity mode. In addition, unlike the bulk emission, the emission from the cavity is highly polarized. The inset to Fig. 2a shows the spectra taken at polarization directions that are parallel and orthogonal with respect to the nanobeam. We obtain an emission polarization that is orthogonal to the nanobeam, in agreement with finite-difference time-domain simulations.

In order to conclusively demonstrate that the nanocavity enhances the spontaneous emission of the nanoplatelets, we perform time-resolved measurements of the cavity emission (see Methods). Figure 2b shows results for emission from the cavity and from the un-patterned surface. In both cases, the decay is bi-exponential, consistent with previous reports[36]. The fast decay component corresponds to direct radiative recombination of excitons in the nanoplatelet, and recent work has shown that the slow decay is likely due to other radiative decay channels involving localized states[37]. The constant background at long timescales (greater than 25 ns) is due to dark counts and collection of residual background photons. Figure 2c shows a histogram of the decay times of the dominant fast decay rate, observed from 10

different devices. For each device, we measure both the cavity decay rate and the decay rate of the nearby un-patterned surface as a reference. The average lifetime of the un-patterned surface is $2.32 \pm 0.22$ ns, in good agreement with previous measurements. The cavity-coupled nanoplatelets exhibit a reduced lifetime of $0.74 \pm 0.14$ ns. We define the Purcell factor $F$ as the ratio of these lifetimes. Figure 2d shows a histogram of $F$ obtained from all the devices. The average value of the Purcell factor is $\overline{F} = 3.1 \pm 0.3$. A similar analysis for the slow decay rate yields a Purcell factor of $\overline{F} = 2.7 \pm 0.3$, which supports the interpretation that both channels correspond to radiative decay processes[37].

We compare the measured Purcell factor to the theoretically predicted maximum value, given by[38]

$$F_{\max} = 1 + \frac{3\lambda^3}{4\pi^2 n^2} \frac{Q_{np}}{V} \psi(\mathbf{r}) \tag{1}$$

The above expression corresponds to the Purcell factor for an emitter located on the nanobeam surface at the location of maximum field intensity. Here, $\lambda$ is the cavity-mode wavelength, $Q_{np} = \lambda/\Delta\lambda_{np}$ is the quality factor of the nanoplatelet emission line with linewidth $\Delta\lambda_{np}$, $n$ is the refractive index of the dielectric material that makes up the cavity, $V = \int d^3\mathbf{r}\varepsilon(\mathbf{r})|E(\mathbf{r})|^2 / [\varepsilon(\mathbf{r})|E(\mathbf{r})|^2]_{\max}$ is the cavity mode volume, $\varepsilon(r)$ is the relative dielectric constant at the cavity frequency, and $\psi(\mathbf{r}) = |E(\mathbf{r})|^2/|E(\mathbf{r})|^2_{\max}$ is the ratio of cavity-field intensity at location $\mathbf{r}$ to the maximum field intensity. We note that the above equation is different from the conventional expression for the spontaneous emission rate enhancement factor, which depends on the ratio of the cavity quality factor $Q$ and the mode volume $V$. The conventional expression applies only in the limit where the linewidth of the emitter is much narrower

than the cavity linewidth. Our system operates in the opposite regime, where the linewidth of the emitter is much broader than the cavity. In this regime, the correct expression for the Purcell factor is given by Eq. 1.[38] From finite-difference time-domain simulations, we obtain $V = 0.55(\lambda/n)^3$ and $\psi(r) = 0.36$ for an emitter located on the nanobeam surface at the field maximum. Substituting these values into Eq. 1, we obtain a maximum Purcell factor of 4.1. Thus, although the field is lower at the surface as compared to the centre of the slab, the surface still provides a sufficiently large local density of states to attain a high coupling efficiency. We attribute the lower Purcell factors measured in the actual devices, as compared to the calculations, to the fact that not all of the nanoplatelets are located at the field maximum.

The measured Purcell factor provides a lower bound on the fraction of spontaneous emission coupled into the cavity mode, which we denote as $\beta$. Using the relation $\beta > 1 - 1/F$ (see Supplementary Note 1), we obtain a lower bound of $\beta > 0.68$. This high coupling efficiency suggests that the devices should be able to support low-threshold lasing.

**Characterizing lasing**. To investigate lasing, we measure the cavity output as a function of excitation power. We plot the resulting spectra in Fig. 3a (for a different device than the one discussed above), along with Lorentzian fits. We use a longer integration time for the lower pump power spectra in order to obtain enough signal to overcome the detector noise. Figure 3b shows the measured intensity and linewidth, determined from the Lorentzian fit, as a function of the total incident power. For the intensity, we normalize the area under the different spectra by the integration time. The intensity plot exhibits a soft threshold around a pump power of 1 μW, which is characteristic of a laser with high $\beta$[39]. Using the calculated spot size of 0.38 μm in radius (see Methods), this pump power corresponds to an intensity of 220 W cm$^{-2}$. The solid line is a fit to a rate-equation model (see Supplementary Note 2)[39] given by

$$P_{in} = \frac{\hbar\omega_p\gamma}{\beta\eta_{in}} \left[ \frac{p}{1+p}(1+\xi)(1+\beta p) - \xi\beta p \right] \qquad (2)$$

where $P_{in}$ is the pump input power, $\gamma$ is the cavity decay rate, $\omega_p$ is the pump frequency, $\xi$ is the cavity photon number at transparency, and $\eta_{in}$ is the pumping efficiency, defined as the fraction of incident pump power absorbed by the laser. We define the cavity photon number as $p = P_{out}/\hbar\omega\gamma\eta_{out}$, where $P_{out}$ is the measured output power, $\omega$ is the cavity resonance frequency, and $\eta_{out}$ is the laser output collection efficiency. Figure 3b shows the best fit to the model in Eq. 2 as a solid line (see Supplementary Note 2). From the fit, we determine a spontaneous coupling efficiency of $\beta = 0.81 \pm 0.03$, which is consistent with the lower bound determined from the lifetime measurements. The figure also shows, for comparison, theoretical curves for other values of $\beta$.

To show the threshold more clearly, we plot the difference between the output intensity and a linear fit to the data for input powers below threshold (Fig. 3c). A device with no threshold will have the same slope both above and below threshold, resulting in a horizontal line. In contrast, a device with a threshold will exhibit a rise in power above threshold. The measured data show a clear nonlinear behaviour with an upward inflection point at the threshold. We also plot theoretical curves for several values of $\beta$.

The emission linewidth (Fig. 3b bottom) provides additional strong evidence of lasing. Below threshold, the linewidth decreases with increasing pump power owing to absorption saturation and reduced spontaneous-emission noise. At threshold

(indicated by the *vertical dashed line*), we observe linewidth broadening due to gain-index coupling, indicating the dynamic phase transition into lasing[40]. Based on the linewidth slightly below this threshold pump power, we estimate the bare cavity quality factor to be 5600. Above threshold, the linewidth again decreases, owing to continued suppression of spontaneous emission noise. This characteristic linewidth behaviour provides a clear signature that our device reaches and exceeds the lasing threshold to attain laser oscillation.

We determine the lasing threshold power from the condition where there is on average one photon in the cavity ($p = 1$)[39]. Inserting this condition into Eq. 2 and using values from the numerical fit, we obtain a lasing threshold power of $P_{th} = P_{in}|_{p=1} = 0.97 \pm 0.03$ μW. This threshold power accounts for all of the pump light injected into the focusing lens. Many works also report the amount of pump light absorbed by the gain material at threshold. We determine a precise value for the absorbed power using the in-coupling efficiency of $\eta_{in} = 21.6 \pm 0.4\%$ obtained from the numerical fit (see Supplementary Note 2). This efficiency is defined as the fraction of the pump laser that is absorbed by the gain material. Multiplying this value by the threshold pump power, we calculate a threshold absorbed power of $210 \pm 10$ nW. This exceptionally low power can be attributed to the high gain provided by the nanoplatelets owing to reduced Auger recombination rates compared to conventional colloidal nanocrystals[19], the high optical absorption coefficients and the absence of inhomogeneous broadening of the nanoplatelets, and the small mode volume of the photonic-crystal cavity. Reducing the mode volume leads to an enhancement of the radiative recombination rate within the nanoplatelets; this, in turn, leads to a large fraction of spontaneous emission being coupled into the cavity. In addition, the spontaneous emission enhancement means that the nanoplatelets can support higher carrier densities before Auger recombination begins to compete with the gain[41].

We also examined the photostability of our device by pumping it continuously with a pump power of 10 μW (approximately 10 times above threshold). The measured output power from the device decreases by only 5% over 2 h of operation (see Supplementary Fig. 1 and Supplementary Note 3). This stability could be further improved by incorporating better surface passivation methods or encapsulating layers[42].

**Discussion**

In summary, we have demonstrated a nanolaser using solution-processable semiconductor nanocrystals that supports room temperature continuous-wave operation. We attain a lasing threshold of only 0.97 μW. To put this threshold value into context, we compare it to previously reported threshold values for nanolasers. Currently, epitaxially grown III–V semiconductor heterostructures represent some of the most advanced nanolaser material systems. A number of works reported room temperature nanolasers with continuous-wave operation based on these materials when coupled to dielectric[24] and metallic[25] cavities. To properly compare our thresholds to these past works, we first note that multiple definitions of laser threshold exist in the literature. The most common definition uses a linear fit of the of the light-in light-out curve above threshold, and defines threshold as the intersection of this fit with the x-axis[1, 3, 43]. Using this definition, room-temperature thresholds as low as 300 nW have been reported in similar nanobeam cavities using InAs quantum dots[44]. We calculate the threshold of our device based on this definition to be 200 nW. However, this definition will significantly under-estimate the threshold when applied to a high-$\beta$ laser such as the one reported in ref. [38] (the reported

$\beta$ is 0.88), as well as in this current work. The more fundamental definition, originally proposed by Björk and Yamamoto[39], identifies threshold as the pump power where the average cavity photon number is unity. Several works use this definition of threshold[30, 45, 46], with reported values as low as $7\,\mu W$. The threshold power we report of $0.97\,\mu W$ is based on this definition and represents, to the best of our knowledge, the lowest threshold input power reported when using the Björk and Yamamoto definition. Finally, some works report the total absorbed power in the gain medium, rather than the input power. The absorbed power accounts only for the energy absorbed in the gain medium, after normalizing out imperfect overlap of the pump laser spot with the active region of the nanolaser or partial transparency of the gain medium at the pump wavelength. Our absorbed pump power is $210\,nW$, similar to previously reported values using indium phosphide structures that achieved sub-microwatt threshold absorbed powers[30].

We attribute the low thresholds and continuous-wave operation attained in this work to a combination of the high radiative efficiency of heterostructure colloidal nanoplatelets at room temperature, along with the reduction in threshold achieved by employing high-Q nanocavities. We note that the heterostructure nanoplatelets can still exhibit blinking behaviour that strongly depends on their environment and pumping intensity. However, since the nanolaser is composed of many nanoplatelets, we expect this blinking to average out, resulting in steady continuous wave emission. Also, these cavities attain high spontaneous emission coupling efficiencies and significantly reduce the volume of the gain medium. These factors serve to significantly reduce the gain required to achieve threshold as compared to previous work based on amplified spontaneous emission[17, 18], which requires larger gain materials and couples only a small fraction of spontaneous emission into the amplified modes.

We could further reduce the laser threshold by improving the cavity Q, which has the potential to exceed 80,000 with better fabrication[47]. Nanoplatelets also provide broad flexibility with respect to the emission wavelength. By synthesizing nanoplatelets with different thicknesses and nanoplatelet heterostructures, we can tune the emission of colloidal nanoplatelets to span the entire visible range[18]. Electrical pumping methods demonstrated in colloidal quantum dot light-emitting diodes[48] may also translate to nanoplatelets to enable electrically pumped nanolasers. Ultimately, colloidal nanoplatelets provide a promising new approach for colloidally synthesized nanophotonic devices operating at extremely low powers.

## Methods

**Cavity design and fabrication**. We adopt a nanobeam photonic-crystal cavity design[34]. The cavity consists of a 200-nm-thick and 300-nm-wide silicon nitride beam with a one-dimensional periodic array of air holes ($a = 250$ nm and $r = 70$ nm). The cavity is formed by linearly reducing the lattice constant from 250 to 205 nm and the hole radius from 70 to 55 nm over a span of 4 holes from both sides of the centre of the beam. This design results in a cavity with a resonance wavelength of 646 nm, and a calculated quality factor $Q = 1.1 \times 10^6$, obtained using numerical finite-difference time-domain simulations that assume perfect fabrication.

The fabrication of the cavity is performed by first depositing 200 nm of stoichiometric silicon nitride on silicon using low-pressure chemical vapour deposition. We pattern the nanobeam photonic crystal cavities using electron-beam lithography and fluorine-based inductively coupled plasma dry etching. We then wet etch the underlying silicon by aqueous KOH to create a suspended nanobeam. Finally, we drop-cast a 10 nM nanoplatelet solution in a 9:1 mixture of hexane and octane onto the sample. From scanning electron microscope images of the device after deposition, we ascertain that the nanoplatelets are uniformly deposited on the surface of the devices (see Supplementary Fig. 2), which is confirmed by consistent photoluminescence emission intensity when exciting different regions on the sample.

**Nanoplatelet synthesis and characterization**. We grew CdSe nanoplatelet cores using a slightly modified version of the recipe by S. Ithurria, from the work

reported by M. Pelton et al.[49]. In a three-necked flask, we degassed 170 mg of cadmium myristate in 15 ml of 1-octadecene (ODE) at 80 °C for 15 min, and then cooled down the solution to room temperature. In a nitrogen glove box (GB) we weighed out 12 mg of Se powder and added it to the degassed solution outside the GB. We then degassed the resulting mixture (cadmium myristate and Se in ODE) at 90 °C for 30 min. After that, we heated up the mixture rapidly, at a rate of 18–20 °C per min. In the meantime, we weighed out 40 mg of freshly ground Cd acetate dehydrate and added it to the solution at the instant when the temperature reached 190 °C. The mixture was still heated up quickly until 220 °C, whereupon we lowered the heating rate to let the mixture reach 240 °C without overshooting. We kept the reaction mixture at 240 °C for 5 min, then we rapidly cooled the mixture down to 150 °C, allowed it to cool down slowly to 70 °C, injected the solution of 2 ml of oleic acid in 10 ml of anhydrous hexane from the GB, and allowed the reaction mixture to cool down to room temperature. In the GB, we collected the reaction mixture in a centrifuge tube and centrifuged it at 5000 rpm for 10 min to isolate the NPLs from QDs and other species in the solution. We discarded the supernatant, collected the precipitate, dispersed the latter in 4 mL of anhydrous hexane, and filtered the solution through a 0.2 μm PFTE filter. The resulting CdSe platelets luminesce at a wavelength of 512 nm, corresponding to a nominal nanoplatelet thickness of four monolayers.

We grew CdS shells around the CdSe cores described above by using one of the variants of the colloidal atomic layer deposition titled "c-ALD Growth in Polar Phase (e.g., NPLs). Method C" in the work by S. Ithurria and D. V. Talapin[50], with some modifications. All manipulations were performed inside the GB using anhydrous hexane, ethanol, acetonitrile, toluene, and dried N-methylformamide (NMF).

Before the growth of the first layer of CdS on the surface of CdSe NPLs, we precipitated CdSe NPLs out of the stock solution with ethanol (the amount varied between 2 and 2.5 ml), centrifuged the resulting suspension at 9000 rpm for 2 min, discarded the solution and re-dispersed the precipitate in 4 ml of hexane. We performed two such washing cycles to remove excess cadmium acetate from the synthesis.

To the washed solution, we added 1 ml of NMF, introduced 50 μL of 40–48% aqueous ammonium sulfide to the NMF layer, and stirred the mixture for 1 min. After the complete phase transfer from hexane to NMF, we discarded the hexane layer, added 4 ml of pure hexane, stirred the mixture, and discarded the hexane layer again. Our next step was to remove excess $S^{2-}$ from the NMF solution of thus formed S-capped NPLs, in order to prevent secondary nucleation of CdS in the next step. To the solution, we added 1.5 ml of acetonitrile and 1 ml of toluene to precipitate the NPLs. We centrifuged the solution at 3800 rpm for 3 min, discarded the solution, and dispersed the precipitate in 1 ml of NMF. This washing procedure was repeated a second time. After the second washing, we re-dispersed the solution in 0.25 mL of NMF, introduced 1.75 mL of 0.2 M cadmium acetate in NMF, and stirred the solution for 1 min. Then, we precipitated the NPLs with 4 ml toluene, centrifuged them, and re-dispersed the precipitate in 1 ml of NMF. To the solution, we added 4 ml hexane and 200 μl of dried 70% technical-grade oleylamine. We stirred the mixture for 1 min and centrifuged it to complete the phase separation. Secondary nucleates of CdS stayed in the NMF. We collected the upper hexane layer with oleylamine-capped CdSe@CdS NPLs – the product of one cycle of the shell growth. We repeated the process of shell growth until a 4-monolayer-thick CdS shell was grown on the CdSe cores.

We deposited the CdSe core/CdS shell nanoplatelets on our nanobeam cavity sample by drop-casting. Supplementary Fig. 2 shows a scanning electron microscope image of the sample after drop-casting, which shows that the nanobeam cavity is coated with a uniform layer of nanoplatelets. We note that the image does not resolve individual nanoplatelets due to the resolution limit of the SEM, and the exact thickness of the film is unknown. However, we also corroborate the uniformity of the deposition by optically exciting different unpatterned regions of the sample and observing very similar emission intensities.

**Optical measurements**. The sample was loaded into a sealed chamber filled with purified nitrogen gas to avoid oxidation and photo-bleaching of nanoplatelets. Sample excitation and collection were performed by a confocal microscopy system using an objective lens with a numerical aperture of 0.6. The cavity emission signal was filtered out spectrally by a 600 nm long-pass filter and spatially using a pinhole aperture. We performed polarization analysis using a half-wave plate and a polarizing beam-splitter.

**Emission spectrum and CW lasing characteristics measurements**. We used a continuous-wave 532.8-nm laser diode as an excitation source. We use the Gaussian beam formula given by $w = (\lambda f/\pi w_0)$ to estimate the pump spot size. In our experiment, the pump fills only 75% of the input aperture, thus $(w_0/f) = 0.75 \times NA$. By substituting these values into the above formula, we calculate a laser spot size of 0.38 μm in radius. The signal collected by the objective lens is focused into a grating spectrometer with a nitrogen-cooled CCD camera at one exit port to acquire the spectrum.

**Time-resolved lifetime measurements**. We pump the sample using laser pulses with 70-ps duration at a wavelength of 405 nm. The laser pulse repetition

rate is 20 MHz, and we set the average power to 1 μW. The collected signal is focused into a grating spectrometer for spectral filtering, and we detect the filtered signal using avalanche photodiode single-photon counters. A time interval analyzer correlates the detected photon events with the pump laser pulses to attain a lifetime trace.

**Data availability**. The authors declare that all data supporting the findings of this study are available within the article and its Supplementary Information files, or are available from the authors upon request.

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

## Acknowledgements

The authors acknowledge support from an Office of Naval Research ONR grant (award number N000141410612), a Defense Advanced Research Projects Agency (DARPA) Defense Science Office grant, the Air Force Office of Scientific Research (AFOSR) (award number FA9550-14-1-0367), the National Science Foundation (award number) DMR-1629601, and the University of Chicago NSF MRSEC Program (award number DMR-14-20703).

## Author contributions

Z.Y., E.W., and M.P. conceived and designed the experiment and prepared the manuscript. Z.Y. fabricated the devices, carried out the measurements, and analyzed the data. E.W., M.P., and Z.Y. carried out the theoretical analysis. I.F. and D.V.T. synthesized nanoplatelet samples. All authors contributed to discussions.

## Additional information

**Competing interests:** The authors declare no competing financial interests.

