## [Peer Review File · Nature Communications]

Reviewers' comments:

Reviewer #1 (Remarks to the Author):

Colloidal nano-crystals have been used for nanolasers for this paper, and is a promising for silicon photonics. The results is very good and will influence the field, as a very low lasing threshold has been demonstrated with high-Q cavity. However, this result in incremental, not ground breaking.

Reviewer #2 (Remarks to the Author):

This manuscript reports a nanolaser that achieves low threshold continuous wave (CW) lasing at room temperature using colloidal CdSe/CdS nanoplatelets that are incorporated into SiN-based nanobeam cavity. Recently, nanoplatelets have proven to be efficient optical gain media (Refs. 11-14). Also, nanobeam cavities have shown ultralow lasing thresholds thanks to the achievement of high spontaneous emission coupling factors. To date, to my knowledge, efficient colloidal nanocrystal gain media such as nanoplatelets have not been considered for nanolasers to date.

Here, a room temperature CW pumped lasing has been achieved thanks to the strong and efficient photonic coupling between the nanocavity and the nanoplatelets. Small mode volume of the cavity enabled the achievement of lasing at very low incident power levels while optical gain is efficiently provided by the core/shell nanoplatelets that have been previously shown to suppress competing nonradiative recombination channels (e.g., Auger recombination) (Ref. 23 and many others) and achieve large gain coefficients (Ref. 11 and 13).

CW pumped lasing has been a long-standing goal in solution-processed gain media. Previously, Ref. 14 has claimed CW pumped optical gain using nanoplatelets with exceptionally low threshold levels ($\sim 6\text{W}/\text{cm}^2$), though any other report could not achieve similar performance levels again. Here, a different strategy is pursued to realize CW pumped lasing in nanoplatelets via employing the nanolaser concept that has been intensively investigated for other materials such as organic semiconductors and epitaxial quantum structures (e.g., quantum wells and quantum dots). Therefore, introducing colloidal nanocrystals into a nano-cavity and the achievement of CW lasing is appealing.

Although I have found the current manuscript interesting, there seem to exist critical technical points that need to be carefully addressed:

First, the claim of the paper on the achievement of "the lowest threshold for any laser operating at room temperature" seems to be wrong. My quick search on similar nanobeam cavity lasers has returned the following work by J. S. Harris and J. Vuckovic et al., (Nanobeam photonic crystal cavity quantum dot laser, *Optics Express* 18, 8781 (2010) <https://www.osapublishing.org/oe/abstract.cfm?uri=oe-18-9-8781>) that achieved better performance level (room temperature CW pumped lasing threshold $\sim 300\text{ nW}$) than the reported threshold (980 nW) of the current manuscript. This previous study employed high-temperature gas-phase epitaxy-grown semiconductor such as InAs quantum dots embedded in GaAs. The threshold seems to be based on the incident power as reported by the current work. Thus, the claims made throughout the manuscript should be revised.

Second, the threshold for lasing is found to be $\sim 1\mu\text{W}$. Therefore, the pump intensity is $\sim 220\text{ W}/\text{cm}^2$ since the spot side radius is reported as 380 nm . Considering the $\sim 1\text{-}2\text{ ns}$ lifetime of the nanoplatelets (as seen from Figure 2c), then the excitation energy dumped into the nanoplatelets during their exciton lifetime is $\sim 200\text{-}400\text{ nJ}/\text{cm}^2$. Previously, optical gain has been shown to be achieved at $\sim 6\text{-}20\text{ }\mu\text{J}/\text{cm}^2$ pump intensity levels (under fs laser pumping) when using the same core/shell nanoplatelets (Refs. 11, 12 and others). Thus, I am not fully convinced that the translated pumping levels could induce enough gain in these nanoplatelets.

Third, in the manuscript it has been claimed that “the development of nano-lasers using solution-processable semiconductor nanocrystals has proved extremely challenging”. However, I cannot see a particular challenge that has been resolved by the technique provided by the current manuscript. The stated challenge, which is Auger recombination, has been there for longtime and being addressed by the colloidal community by using different nanocrystal architectures (e.g., graded shell, thick shell, nanoheterorods, hetero-nanoplatelets, etc.). Current manuscript nicely combines core/shell nanoplatelets, which have been known to be efficient for optical gain, with previously well-studied nanobeam cavities to achieve low threshold lasing under CW pumping. However, I do not see any challenge that has been particularly resolved by the current manuscript. Thus, the novelty of the work sounds weak.

Other points:

- 1) On line 84-85, it has been stated that emission collected at the patterned area (the cavity region) comes from a significantly lower number of platelets. If the pump spot is the same and the coverage of the nanoplatelets is homogeneous, then this claim does not make sense.
- 2) The fast lifetime component of the PL decay has been attributed to the radiative recombination in the nanoplatelets. However, this could be misleading and also may not be correct as show by other recent works (e.g., M. Olutas et al. ACS Nano 9, 5041 (2015) and F. T. Rabouw et al. Nano Letters 15, 7718 (2015)). If amplitude averaged lifetimes were to be used to calculate the Purcell factors, would that create significant difference?
- 3) On line 82, ~8 times brighter emission of the nanoplatelets in the cavity region has been reported as compared to the nanoplatelets outside of the cavity region. Does PL intensities correlate well with the Purcell factor calculated from PL decay kinetics?
- 4) What is the thickness of the nanoplatelet layer? Is it monolayer or sub-monolayer coverage? It would be better if high resolution SEM images could be provided to replace Figure S1.
- 5) Optional: Would it be possible to measure the coherency time of the emitted laser which would make a strong point that the measured emission is actually due to lasing?
- 6) Would it be possible to provide transmission spectrum of the cavity or the spectral positions of the photonic modes of the nanocavity?
- 7) In the introduction, it would be better to compare and contrast nanoplatelets to other epitaxial systems (quantum dots/wells integrated into nanobeam cavities or different nanolasers instead of mentioning quantum wires).

Reviewer #3 (Remarks to the Author):

In “A room temperature continuous-wave nano-laser using colloidal quantum wells”, Yang et al. present the fabrication and optical characterization of a nanolaser consisting of a suspended silicon nitride 1D hole array cavity coated with colloidal nanoplatelets as the gain medium. The manuscript features two primary results. The first is a lasing threshold of 1uW input power, which the authors contend is the lowest reported threshold for any cw laser at room temperature. The second result is cw lasing at room temperature for over 2 hours with minimal decrease in intensity.

While these results are presented as milestones, I have three primary concerns/misgivings about these results and thus, their importance (see below). As such, I cannot recommend this manuscript for publication until these points are clarified.

- 1) The lasing threshold of 1 uW is claimed to be a milestone. However, it is not clear to me why a unit of power, joules/sec, is a meaningful metric for the lasing threshold. Wouldn't a more appropriate metric be an intensity, or excitation flux, in units of W/cm²? Can the authors please explain why uW is a meaningful metric here? Or are the authors trying to highlight that because each device is so small, the total amount of power needed for excitation to lasing is also small?

When the excitation flux is calculated (given the correct excitation spot size, see below), how does this value compare to other optically-pumped lasers? Do these devices still truly have the lowest pump thresholds?

2) The second primary result is cw lasing over two hours. This is an impressive result for a nanocrystal-based laser, yet there is no data showing this. Please provide this data.

3) The data in Fig 3a and 3b are not consistent. For instance, in Fig 3a, the output intensity at 0.1 uW pump is ~400 counts and the output at 1.1 uW pump is ~900 counts. So, according to Fig 3a, an 11x increase in pump intensity below threshold results in a 2.25x increase in output intensity. However, Fig 3b shows that below threshold, a 10x increase in pump intensity results in a ~10x increase in output intensity – the expected linear trend. Can the authors please explain this discrepancy between Fig 3a and 3b?

I would like to see the authors provide explanations for these concerns. I hold these concerns to be serious enough that, until then, I cannot recommend publication of this manuscript in Nature Communications.

Below, I list a few more minor concerns/questions/suggestions for improving the manuscript.

a) The authors mention that “the development of nano-lasers using solution-processable semiconductor nanocrystals has proved extremely challenging”. However, there is no explanation of why this has been the case. In fact, the nanobeam cavity fabrication consists of established methods and the nanoplatelets are deposited simply by dropcasting. It is unclear in what way the development of nanocrystal nanolasers has been hindered and now overcome by this work. Can the authors please elaborate?

b) The authors mention that intersystem crossing into dark triplet states makes cw operation difficult. But don't nanoplatelets also show blinking? Even if it's not a triplet state, they are still non-emissive. How is it then that these materials can show cw operation?

c) Figure 2b: I could be mistaken, but the fit to the PL decay from the nanoplatelets in the cavity appears to be triple exponential, not double exponential as written in the text. I think it would be good to double check.

d) The authors claim that the laser spot size is 0.38 um in radius. However, with a 532 nm laser and a 0.6 NA objective, shouldn't the diffraction-limited spot size be $\sim \lambda/2NA$, which is 0.44 um? Is 0.38 um radius correct?

Reviewer #1

Colloidal nano-crystals have been used for nanolasers for this paper, and is a promising for silicon photonics. The results is very good and will influence the field, as a very low lasing threshold has been demonstrated with high-Q cavity. However, this result in incremental, not ground breaking.

We thank the reviewer for their positive comments regarding our paper, but strongly disagree with their final statement. Our work demonstrates, to the best of our knowledge, the lowest threshold laser using colloidal emitters reported to date. We accomplished this milestone not through incremental improvements on previous devices, but by making a kind of laser that had never previously been built: a colloidal-nanocrystal nano-laser. It is also the first nano-laser based on solution-processed materials, and demonstrates the practicality of building stable, room-temperature, continuous-wave lasers using nanocrystals. Our device is stable over several hours and operates well above threshold, which are also important milestones that have been difficult to achieve using solution-processable materials. The previous publication by some of the co-authors on amplified spontaneous emission using colloidal nanoplatelets has been cited 92 times in the 33 months since it was published, indicating great interest in the community. The current manuscript represents a similar major step.

We are therefore confident that our results are indeed ground-breaking.

Reviewer #2

1. This manuscript reports a nanolaser that achieves low threshold continuous wave (CW) lasing at room temperature using colloidal CdSe/CdS nanoplatelets that are incorporated into SiN-based nanobeam cavity. Recently, nanoplatelets have proven to be efficient optical gain media (Refs. 11-14). Also, nanobeam cavities have shown ultralow lasing thresholds thanks to the achievement of high spontaneous emission coupling factors. To date, to my knowledge, efficient colloidal nanocrystal gain media such as nanoplatelets have not been considered for nanolasers to date.

Here, a room temperature CW pumped lasing has been achieved thanks to the strong and efficient photonic coupling between the nanocavity and the nanoplatelets. Small mode volume of the cavity enabled the achievement of lasing at very low incident power levels while optical gain is efficiently provided by the core/shell nanoplatelets

that have been previously shown to suppress competing nonradiative recombination channels (e.g., Auger recombination) (Ref. 23 and many others) and achieve large gain coefficients (Ref. 11 and 13).

CW pumped lasing has been a long-standing goal in solution-processed gain media. Previously, Ref. 14 has claimed CW pumped optical gain using nanoplatelets with exceptionally low threshold levels ($\sim 6\text{W/cm}^2$), though any other report could not achieve similar performance levels again. Here, a different strategy is pursued to realize CW pumped lasing in nanoplatelets via employing the nanolaser concept that has been intensively investigated for other materials such as organic semiconductors and epitaxial quantum structures (e.g., quantum wells and quantum dots). Therefore, introducing colloidal nanocrystals into a nano-cavity and the achievement of CW lasing is appealing.

We thank the reviewer for their accurate and positive comments and their clear appreciation of the rationale and significance of our work.

2. Although I have found the current manuscript interesting, there seem to exist critical technical points that need to be carefully addressed: First, the claim of the paper on the achievement of “the lowest threshold for any laser operating at room temperature” seems to be wrong. My quick search on similar nanobeam cavity lasers has returned the following work by J. S. Harris and J. Vuckovic et al., (Nanobeam photonic crystal cavity quantum dot laser, Optics Express 18, 8781 (2010) <https://www.osapublishing.org/oe/abstract.cfm?uri=oe-18-9-8781>) that achieved better performance level (room temperature CW pumped lasing threshold $\sim 300\text{ nW}$) than the reported threshold (980 nW) of the current manuscript. This previous study employed high-temperature gas-phase epitaxy-grown semiconductor such as InAs quantum dots embedded in GaAs. The threshold seems to be based on the incident power as reported by the current work. Thus, the claims made throughout the manuscript should be revised.

We thank the reviewer for pointing out this paper which we missed, and which should clearly have been cited. This manuscript does claim to have a threshold of only 300 nW, but this value cannot be compared to the 970 nW threshold we report in the manuscript because it is calculated differently. The authors of this work calculate the threshold of their laser by using a linear fit to the light-in light-out curve (Figure 3 of the manuscript) above threshold and finding the intersection with the x-axis. When we calculate the threshold of our laser using this method, we attain a threshold of 200 nW, which is lower than their attained value. However, this definition of threshold is not strictly correct. It provides a good estimate of the threshold for a low β laser.

But it cannot be used for high β lasers such as the one reported both the manuscript by Harris and Vuckovic ($\beta = 0.88$) and our manuscript ($\beta = 0.81$), and if used it will significantly under-estimate the threshold. The more accurate definition of laser threshold is defined by Björk and Yamamoto [Ref. 33] as the condition where the average number of photons in the cavity is equal to one. This definition works for both low and high β lasers and is therefore more fundamental. It is this definition that we employ here.

In order to avoid any confusion about our claims, we have added discussion in the “Discussion” section that reviews the definitions of threshold for a high β , what has been previously reported in the literature, and compares these past works with what we report in this manuscript. Also, due to the multiple definitions of threshold, which now necessitates a more extensive discussion, we have replaced the statement “the lowest threshold for any laser operating at room temperature” with the statement “any laser using colloidal emitters”. Because our thresholds are orders of magnitude smaller than those reported in other colloidal materials, we can make this statement without regard to the definition of threshold. The discussion at the end of the manuscript then compares the threshold values attained in our device to those obtained in other material systems using the various definitions of threshold, and shows that it is in fact lower.

3. Second, the threshold for lasing is found to be $\sim 1\mu W$. Therefore, the pump intensity is $\sim 220 W/cm^2$ since the spot side radius is reported as 380 nm. Considering the $\sim 1-2$ ns lifetime of the nanoplatelets (as seen from Figure 2c), then the excitation energy dumped into the nanoplatelets during their exciton lifetime is $\sim 200-400 nJ/cm^2$. Previously, optical gain has been shown be achieved at $\sim 6-20 \mu J/cm^2$ pump intensity levels (under fs laser pumping) when using the same core/shell nanoplatelets (Refs. 11, 12 and others). Thus, I am not fully convinced that the translated pumping levels could induce enough gain in these nanoplatelets.

The reviewer is correct that the pump intensity is about $220 W/cm^2$ for our device working at threshold. But one cannot easily compare the gain required to achieve pulsed ASE in a film of nanoplatelets with the gain required to achieve cw lasing in a high-Q cavity. In the film, ASE occurs when the net modal gain is positive; that is, when optical gain for guided light in the film overcomes propagation losses, which are dominated by scattering. In a high-Q cavity, lasing occurs when the intracavity gain overcomes cavity losses. One can therefore achieve lasing in principle with any amount of gain, provided the the quality factor of the cavity is high enough. In the manuscript, we actually calculate the minimum required input power (equal to the

absorbed power at threshold), which is 210 nW. We attain this value from the condition that the input power must sustain on average one photon in the cavity, and this is the appropriate value to compare to the actual pump power to ensure that we have a sufficient pump to attain lasing. Our actual input power is 0.97 μ W, approximately five times larger, so it is quite sufficient to attain lasing. The difference between the actual pump power and the minimum pump power is well explained by the fact that our spot size is larger than the cavity mode volume, and therefore only a fraction of our pump overlaps with the gain medium and efficiently pumps the nanoplatelets. We have added a clarification in the “Discussion” section: “We attribute the low thresholds attained in this work to a combination of the high radiative efficiency of colloidal nanoplatelets at room temperature, along with the reduction in threshold achieved by employing high-Q nano-cavities. These cavities attain high spontaneous emission coupling efficiencies and significantly reduce the size of the gain medium. These factors serve to significantly reduce the gain required to achieve threshold as compared to previous work based on amplified spontaneous emission^{11,12}, which requires larger gain materials and couples only a small fraction of spontaneous emission into the lasing modes.”

4. Third, in the manuscript it has been claimed that “the development of nano-lasers using solution-processable semiconductor nanocrystals has proved extremely challenging”. However, I cannot see a particular challenge that has been resolved by the technique provided by the current manuscript. The stated challenge, which is Auger recombination, has been there for longtime and being addressed by the colloidal community by using different nanocrystal architectures (e.g., graded shell, thick shell, nanoheterorods, hetero-nanoplatelets, etc.). Current manuscript nicely combines core/shell nanoplatelets, which have been known to be efficient for optical gain, with previously well-studied nanobeam cavities to achieve low threshold lasing under CW pumping. However, I do not see any challenge that has been particularly resolved by the current manuscript. Thus, the novelty of the work sounds weak.

Achieving a colloidal nano-laser has been a long-standing goal of many research groups. A number of groups, including our own, previously attempted to use colloidal quantum dots as gain materials in nano-cavities [Ref. 20-23], but none of these works were able to attain lasing. The main problem in all of these past works has been Auger recombination, which destroys the gain. The reviewer is correct that recently a number several groups have reported major breakthroughs in eliminating Auger recombination in quantum dots [Ref. 8-10]. But we are not familiar with any work that integrated these emitters into a nanocavity. We believe there are a number of issues that make such integration challenging: 1) A nano-laser requires a uniform,

dense film of emitters. As we show in Fig. S1 of the supplement, nanoplatelets deposit rather uniformly across the nanobeam. We have had significantly greater difficulty getting uniform films of quantum dots [Ref. 23], and we believe this is quite important to achieve stimulated emission. 2) Material degradation is another major issue when working with thin film samples. With quantum dot samples, we could see clear degradation in a matter of minutes when we used high excitation power, making it extremely difficult to perform any measurements that conclusively demonstrate lasing. Nanoplatelets are much more stable, as we now show in Figure S2. We believe that these reasons make nanoplatelets particularly suitable for nano-laser applications, but we certainly agree that trying to produce these results with quantum dots is an interesting and worthwhile path as well. To clarify the difficulty of achieving nano-lasers with colloidal materials, we have modified the statement in introduction to “A number of works incorporated colloidal quantum dots into nanocavities²¹⁻²³, but were unable to reach lasing due to rapid Auger recombination. Attaining uniform dense films of quantum dots also posed a major challenge, and these films degraded rapidly under optical excitation²⁴. Achieving a low-threshold nano-laser using such materials has therefore remained an outstanding challenge.”

5. On line 84-85, it has been stated that emission collected at the patterned area (the cavity region) comes from a significantly lower number of platelets. If the pump spot is the same and the coverage of the nanoplatelets is homogeneous, then this claim does not make sense.

The size of the nano-cavity mode in our experiment is much smaller than the spot size of the beam. We can approximate the active area by the mode volume divided by the thickness of the membrane which is $0.086 \mu\text{m}^2$, and the spot size of laser is $0.5 \mu\text{m}^2$ (corresponding to a 400-nm diameter spot size). Thus, when we pump the cavity only a small fraction of the spot actually overlaps with nanoplatelets. In contrast, for the un-patterned surface, the entire laser spot overlaps with gain material. Thus, the emission from the cavity corresponds to a much smaller number of platelets than the emission from the un-patterned surface. In order to clarify this point, we have modified the statement to “We observe brighter intensity at the cavity resonance despite the fact that we excite many more nanoplatelets when looking at the bare surface as compared to the cavity. The reason for this disparity is due to the fact that the surface area of the nanocavity is much smaller than the surface area excited by the laser on the unpatterned surface. We calculate the surface area of the cavity, defined as the mode volume multiplied by the membrane thickness, to be $0.086 \mu\text{m}^2$. In contrast, the illuminated surface area on the unpatterned surface is $0.45 \mu\text{m}^2$.”

6. The fast lifetime component of the PL decay has been attributed to the radiative recombination in the nanoplatelets. However, this could be misleading and also may not be correct as show by other recent works (e.g., M. Olutas et al. ACS Nano 9, 5041 (2015) and F. T. Rabouw et al. Nano Letters 15, 7718 (2015)). If amplitude averaged lifetimes were to be used to calculate the Purcell factors, would that create significant difference?

We would like to thank the reviewer bringing up recent works on exciton dynamics of nanoplatelets. These recent publications indicate that multiple radiative decay channels may exist, so it may be better to also examine changes in amplitude-averaged lifetimes, and not just in the shortest lifetime. The amplitude-averaged lifetimes lead to an average Purcell factor of $\bar{F} = 3.0 \pm 0.6$, which is nearly identical to the measured value of $\bar{F} = 3.1 \pm 0.3$ we reported previously, based on the fast decay component. We note the reason for the consistency between two methods comes from the fact that the slower decay, which constitutes about 20% of the whole population, also features an average Purcell factor of $\bar{F} = 2.7 \pm 0.3$. This is consistent with the assertion from these works that both the slow and fast decay channels are radiative. To clarify this point, we have modified the lifetime analysis part to include the rates of enhancement for both decay channels.

7. On line 82, ~8 times brighter emission of the nanoplatelets in the cavity region has been reported as compared to the nanoplatelets outside of the cavity region. Does PL intensities correlate well with the Purcell factor calculated from PL decay kinetics?

It is very difficult to calculate the expected increase in brightness that the reviewer refers to from the Purcell factor alone. This brightness depends on three factors 1) The Purcell factor, 2) The far-field emission pattern of the cavity, and 3) The number of nanoplatelets we pump in the cavity. The first factor affects brightness only above saturation. Below saturation, only factors 2 and 3 determine the brightness, and we do not have a good estimate of these values. Factor #2 depends on the details of the far-field emission pattern and is sensitive to the numerical aperture of the lens. Factor #3 is also difficult to know, because it requires us to quantify the number of nanoplatelets we excite that are sitting in the high field region of the cavity as compared to the number not in the cavity field (i.e. sitting on the cavity mirrors). To clarify this point, we have added the statement “We attribute this increase in brightness to both redirection of the nanoplatelet emission by the cavity as well as the Purcell effect.” in the photoluminescence measurement section.

8. What is the thickness of the nanoplatelet layer? Is it monolayer or sub-monolayer coverage? It would be better if high resolution SEM images could be provided to replace Figure S1.

Unfortunately, the SEM images we show in Figure S1 are the highest resolution images we were able to attain. We were unable to resolve individual nanoplatelets, probably due to the resolution limit of the machine. In addition, it is difficult to perform high-resolution SEM on SiN, which is almost a perfect insulator and is therefore prone to charging and other deleterious effects that degrade the image resolution. We do not currently have access to an imaging method that could resolve the thickness of this layer with precision of a few nanometers. We have added the following sentence in the supplement to clarify this point, “We note that the image does not resolve individual nanoplatelets due to the resolution limit of the SEM. The exact thickness of the film is also unknown.”

9. Optional: Would it be possible to measure the coherency time of the emitted laser which would make a strong point that the measured emission is actually due to lasing?

The coherence time is usually measured using a Michelson interferometer. This measurement is certainly possible, but would not provide additional information, because the coherence time of the emission is just the inverse of the linewidth. The two are directly related by the Fourier transform relation, and basically represent two methods to measure the same thing. Does the reviewer mean the second-order coherence, $g^{(2)}(0)$? This measurement would certainly provide strong evidence of lasing. But the method that we are familiar with to measure it is based on a Hanbury-Brown Twiss experiment, which would require single photon counters with temporal resolution on the order of 3 ps (the coherence time of the laser). We don't have photon counters with this time resolution.

10. Would it be possible to provide transmission spectrum of the cavity or the spectral positions of the photonic modes of the nanocavity?

We agree with the reviewer that a transmission spectrum of the cavity would characterize the cavity mode clearly and enable us to study nonlinear properties. However, this measurement is difficult to perform using a nanobeam structure. Such measurements would require either a cross-polarization reflectivity measurement, which is challenging in nanobeams because the structure scatters a great deal of light, or alternately light injection via an evanescently coupled waveguide, which would require a more complicated device structure. For these reasons, we rely instead on

fluorescence measurements, which are easier to perform using the available device and lasers.

11. In the introduction, it would be better to compare and contrast nanoplatelets to other epitaxial systems (quantum dots/wells integrated into nanobeam cavities or different nanolasers instead of mentioning quantum wires).

We thank the reviewer for their suggestions. We have implemented the reviewer's suggestion by adding a comparison between our colloidal nanoplatelets system. We have performed this comparison in the "Discussion" section instead of the introduction, because it requires a technical discussion that would be more appropriate in this section.

Reviewer 3

1. In "A room temperature continuous-wave nano-laser using colloidal quantum wells", Yang et al. present the fabrication and optical characterization of a nanolaser consisting of a suspended silicon nitride 1D hole array cavity coated with colloidal nanoplatelets as the gain medium. The manuscript features two primary results. The first is a lasing threshold of 1uW input power, which the authors contend is the lowest reported threshold for any cw laser at room temperature. The second result is cw lasing at room temperature for over 2 hours with minimal decrease in intensity.

While these results are presented as milestones, I have three primary concerns/misgivings about these results and thus, their importance (see below). As such, I cannot recommend this manuscript for publication until these points are clarified.

We thank the reviewer for their interest in our work and for their helpful comments. We have modified the manuscript to address all reviewer comments and provide a detailed explanation of these modifications in the discussion below.

2. The lasing threshold of 1 uW is claimed to be a milestone. However, it is not clear to me why a unit of power, joules/sec, is a meaningful metric for the lasing threshold. Wouldn't a more appropriate metric be an intensity, or excitation flux, in units of W/cm²? Can the authors please explain why uW is a meaningful metric here? Or are the authors trying to highlight that because each device is so small, the total amount of power needed for excitation to lasing is also small? When the excitation flux is calculated (given the correct excitation spot size, see below), how

does this value compare to other optically-pumped lasers? Do these devices still truly have the lowest pump thresholds?

The input threshold power is an important metric because it tells how much power one needs to provide in order to operate the laser. This metric is of central interest when engineering devices that should operate at low energies, such as a nano-laser, and is the figure of merit used in virtually all works on nano-lasers (Ref. [1, 3, 4, 18, 24, 37]). We agree with the reviewer that other communities, particularly those working on amplified spontaneous emission, use different metrics for their lasers. In amplified spontaneous emission, there is a tradeoff between the threshold power and output power. Pumping a larger area leads to a larger threshold but also a larger output power, so it is sensible to compare the pump intensity to the output intensity. But in a three-dimensionally confined cavity, and particularly in nano-cavities, such a comparison is not meaningful, because the size of the gain medium is determined by the volume of the cavity mode, not by the spot size; in our experiments, the spot size is much bigger than the cavity mode area (see our response to reviewer 2, comment 5). Doubling the spot size would increase the threshold by a factor of two, but would not increase the output power.

The reduction of threshold in a nano-laser is due to three factors: 1) Reduced volume of the laser, 2) high quality factor of the nano-cavity, and 3) high β due to the Purcell effect. We do not believe it is possible to get the low threshold powers we attain here without these three factors. Specifically, we are not familiar with any work that achieves such low threshold powers using amplified spontaneous emission by pumping only a very small volume, or by any other means that does not make use of a high-Q, three-dimensionally confined nano-cavity. For easy comparison, we also quote the threshold pump intensity of our device in the manuscript.

3. The second primary result is cw lasing over two hours. This is an impressive result for a nanocrystal-based laser, yet there is no data showing this. Please provide this data.

Per the reviewer's suggestion, we have included data and discussion of photostability in the Supplementary materials.

4. The data in Fig 3a and 3b are not consistent. For instance, in Fig 3a, the output intensity at 0.1 uW pump is ~400 counts and the output at 1.1 uW pump is ~900 counts. So, according to Fig 3a, an 11x increase in pump intensity below threshold results in a 2.25x increase in output intensity. However, Fig 3b shows that below threshold, a 10x increase in pump intensity results in a ~10x increase in output intensity – the expected linear trend. Can the authors please explain this discrepancy

between Fig 3a and 3b? I would like to see the authors provide explanations for these concerns. I hold these concerns to be serious enough that, until then, I cannot recommend publication of this manuscript in Nature Communications.

Below, I list a few more minor concerns/questions/suggestions for improving the manuscript.

We thank the reviewer for pointing out this subtlety, which we should have explained better in the manuscript. The spectra plotted in Fig. 3a were obtained using different integration times. For 100 nW pump powers, we integrated for 150 s, whereas we integrated for only 40 s to obtain the spectrum at 1.1 μ W. We required longer integration times at lower pump powers in order to get a signal that sufficiently exceeds the noise of the spectrometer detector. In Fig. 3b, we plot the output intensity, which is the area under each curve normalized by the integration time. We have added a clear explanation of this point in the manuscript.

5. The authors mention that “the development of nano-lasers using solution-processable semiconductor nanocrystals has proved extremely challenging”. However, there is no explanation of why this has been the case. In fact, the nanobeam cavity fabrication consists of established methods and the nanoplatelets are deposited simply by dropcasting. It is unclear in what way the development of nanocrystal nanolasers has been hindered and now overcome by this work. Can the authors please elaborate?

Achieving a colloidal nano-laser has been a long-standing goal of many research groups. A number of groups, including our own, previously attempted to use colloidal quantum dots as gain materials in nano-cavities [Ref. 20-23], but none of these works were able to attain lasing. A central problem in all of these past works has been Auger recombination, which competes with gain. The reviewer is correct that recently several groups have reported major breakthroughs in reducing Auger recombination rates in quantum dots [Ref. 8-10]. However, we are not familiar with any work that integrated these emitters into a nanocavity. We believe there are a number of issues that make such integration challenging: 1) A nano-laser requires a uniform dense film of emitters. As we show in Fig. S1 of the supplement, nanoplatelets deposit rather uniformly across the nanobeam. We have had significantly greater difficulty obtaining uniform films of quantum dots on nanocavities [Ref. 23]. 2) Material degradation is another major issue when working with thin-film samples. With quantum-dot samples, we could see clear degradation in a matter of minutes at high excitation powers, making it extremely difficult to perform any measurements that conclusively demonstrate lasing. Nanoplatelets are much more stable, as we now show in Figure S2.

We believe that these reasons make nanoplatelets particularly suitable for nano-laser applications, but we certainly agree that trying to produce these results with quantum dots is an interesting and worthwhile path as well. To clarify the difficulty of achieving nano-lasers with colloidal materials, we have modified the statement in the introduction to “A number of works incorporated colloidal quantum dots into nanocavities²¹⁻²³, but were unable to reach lasing due to rapid Auger recombination. Attaining uniform dense films of quantum dots also posed a major challenge, and these films degraded rapidly under optical excitation²⁴. Achieving a low-threshold nano-laser using such materials has therefore remained an outstanding challenge.”

6. The authors mention that intersystem crossing into dark triplet states makes cw operation difficult. But don't nanoplatelets also show blinking? Even if it's not a triplet state, they are still non-emissive. How is it then that these materials can show cw operation?

The core/shell heterostructure nanoplatelet nanocrystals we utilized exhibit very little emission intermittency (blinking), as shown in Ref. 27 of the revised manuscript. In addition, the active material in our laser consists of many nanoplatelets, not just one, and the blinking behavior of different nano-platelets is statistically independent. Thus, even though individual nanoplatelets blink “on” and “off,” the total intensity emitted by all of the platelets that are coupled to the cavity does not change significantly. Finally, we note that the Purcell effect is expected to further reduce the effects of blinking for nanoplatelets that are coupled to the laser cavity. We have added the following comments to the manuscript to address these points: “The core/shell structure significantly reduces emission intermittency (blinking) at a single nanoparticle level²⁸. Thus, they can provide higher gain at equivalent excitation powers.” And “We note that the heterostructure nanoplatelets can still exhibit blinking behavior that strongly depends on their environment and pumping intensity. Because the nanolaser is composed of many nanoplatelets, we expect this blinking behavior to average out resulting in a steady continuous wave emission.”

7. Figure 2b: I could be mistaken, but the fit to the PL decay from the nanoplatelets in the cavity appears to be triple exponential, not double exponential as written in the text. I think it would be good to double check.

We believe the reviewer is referring to the flattening of the free-induction decay around 25 ns. We do not believe this flattening is a third exponential decay, but rather that it is due to background and dark-counts. We confirm this by estimating the background level from the data for negative time delays and compare it to the cavity decay signal after 25 ns. The difference of the two signal levels is within a few

percent. We have added a clarification in the manuscript explaining this point, which states “The constant background at long timescales (greater than 25 ns) is due to dark counts and collection of residual background photons.”

8. *The authors claim that the laser spot size is 0.38 μm in radius. However, with a 532 nm laser and a 0.6 NA objective, shouldn't the diffraction-limited spot size be $\sim\lambda/2NA$, which is 0.44 μm ? Is 0.38 μm radius correct?*

We believe the discrepancy between the numbers reported by the reviewer and us is due to a slightly different definition of spot size. In our manuscript, we used the

Gaussian beam formula given by $w = \frac{\lambda f}{\pi w_0}$. In our experiment, the pump only fills

75% of the input aperture, thus $\frac{w_0}{f} = 0.75 \times \text{NA}$. Substituting these values into the

above formula, we calculate a beam waist of 0.38 μm . To clarify the estimation of beam waist, we have included this whole calculation procedure in the Methods section.

Reviewers' comments:

Reviewer #2 (Remarks to the Author):

The authors have satisfactorily answered the questions/concerns raised by this reviewer and revised the manuscript accordingly. At this stage, I have few additional comments:

1) The authors mentioned about the lower limit of the lasing threshold - 210 nW in the response letter but it is not clear to me how this is reached or estimated?

2) The authors should quantitatively compare the cavity loss of their nanobeam cavity to the thin-film scattering loss (or waveguide loss) of conventional nanocrystals films that are used in ASE experiments.

3) There have been several previous reports that coupled colloidal quantum dots into nanocavities (not necessarily nanobeam cavities but many others). These previous works might be contrasted, if any is overlooked in the current manuscript.

4) The authors describe the difficult of measurement of the cavity transmission but suggest to consider cavity modulated photoluminescence spectrum. However, it is not clear whether they provide a new experimental data (except the original data in Fig. 2a) for this in the revised manuscript or not.

Reviewer #3 (Remarks to the Author):

I am satisfied with the reviewer response and the changes made to the manuscript. I think the authors did a commendable job clarifying their statements, qualifying their claims, and discussing their work in the context of the literature. I now recommend this paper for publication in Nature Communications.

Reviewer #2

The authors have satisfactorily answered the questions/concerns raised by this reviewer and revised the manuscript accordingly. At this stage, I have few additional comments:

1. The authors mentioned about the lower limit of the lasing threshold - 210 nW in the response letter but it is not clear to me how this is reached or estimated?

We believe the reviewer is referring to our response to comment 3 from the previous review. In that comment, we mention that the minimum power required to achieve lasing is 210 nW, when considering only the power absorbed by the gain material. We attain this value by numerically fitting the data in Fig. 3b to the theoretical model described in Eq. 2. This model includes the pump efficiency η_{in} , which is defined as the fraction of the pump laser that is absorbed by the gain material. We treat this variable as a fitting parameter. Once we determine η_{in} from the fit, we can multiply it by the actual pump power to extract the total absorbed power. Therefore, the 210 nW is not directly measured, it is extracted by using a numerical fit of the data to a theoretical model. We have added a brief discussion in the text to clarify this point.

2) The authors should quantitatively compare the cavity loss of their nanobeam cavity to the thin-film scattering loss (or waveguide loss) of conventional nanocrystals films that are used in ASE experiments.

We believe the reviewer is again referring to our response to comment 3 from the previous review. There, we said that ASE occurs in films when optical gain for guided light in the film overcomes propagation losses, and lasing occurs in high-Q cavities when the intracavity gain overcomes cavity losses; this has presumably motivated the new comment. However, cavity loss and waveguide loss are different quantities that cannot be directly compared. In a cavity, the losses are given by a field decay rate; in a waveguide, they are given by the absorption length. Moreover, there are no available measurements for the waveguide loss in the films, because these numbers were not reported in the earlier papers on ASE. One indication that these losses are significant is that ASE thresholds varied by about a factor of four from one measurement to another, which presumably is due to variations in the loss. It is therefore not possible to attempt to compare the cavity and waveguide losses quantitatively.

3) There have been several previous reports that coupled colloidal quantum dots into nanocavities (not necessarily nanobeam cavities but many others). These previous works might be contrasted, if any is overlooked in the current manuscript.

Per the reviewer's suggestions, we have added several more citations for coupling colloidal quantum dots to different types of cavities including Fabry–Pérot, distributed Bragg reflector, and microsphere resonators. All of these past works required large pump powers in the milliWatt to kiloWatt range to achieve lasing threshold due to their large cavity mode volume and high loss compared with our device.

4) The authors describe the difficulty of measurement of the cavity transmission but suggest to consider cavity modulated photoluminescence spectrum. However, it is not clear whether they provide a new experimental data (except the original data in Fig. 2a) for this in the revised manuscript or not.

We believe the reviewer is referring to our response to comment 10 from the previous review. In our response, we indicated that cavity transmission measurements are challenging in the nanobeam structure we used, which is why we rely on fluorescence measurements instead throughout our manuscript. The comment was intended to explain why all of our measurements (including Fig. 2a) were performed using the cavity fluorescence rather than the cavity transmission. We did not add any new data, and did not mean to imply that we had. We apologize if there was any confusion.

REVIEWERS' COMMENTS:

Reviewer #2 (Remarks to the Author):

The authors have satisfactorily responded to the concerns/comments of this reviewer. I would recommend the paper for publication.